# A Multi-Center Study for the Development of the Taiwan Cognition Questionnaire (TCQ) in Major Depressive Disorder

**DOI:** 10.3390/jpm12030359

**Published:** 2022-02-26

**Authors:** Yung-Chieh Yen, Nan-Ying Chiu, Tzung-Jeng Hwang, Tung-Ping Su, Yen-Kuang Yang, Cheng-Sheng Chen, Cheng-Ta Li, Kuan-Pin Su, Te-Jen Lai, Chia-Ming Chang

**Affiliations:** 1Department of Psychiatry, E-Da Hospital, Kaohsiung 824, Taiwan; ed103750@edah.org.tw or; 2School of Medicine, I-Shou University, Kaohsiung 824, Taiwan; 3Center for Sleep Medicine, Department of Psychiatry, Chang-Hua Christian Medical System, Chang-Hua 500, Taiwan; 400786@cch.org.tw; 4Lukang Christian Hospital, Chang-Hua 505, Taiwan; 5Department of Psychiatry, National Taiwan University Hospital, Taipei 100, Taiwan; tjhwang@ntu.edu.tw; 6School of Medicine, National Taiwan University, Taipei 100, Taiwan; 7Department of Psychiatry, Taipei Veterans General Hospital, Taipei 112, Taiwan; tpsu@vghtpe.gov.tw; 8Institute of Philosophy of Mind and Cognition, National Yang Ming Chiao Tung University, Taipei 112, Taiwan; 9Department of Psychiatry, Faculty of Medicine, National Yang Ming Chiao Tung University, Taipei 112, Taiwan; 10Division of Psychiatry, School of Medicine, National Yang Ming Chiao Tung University, Taipei 112, Taiwan; 11Department of Psychiatry, General Cheng Hsin Hospital, Taipei 112, Taiwan; 12Department of Psychiatry, National Cheng Kung University Hospital, Tainan 701, Taiwan; ykyang@mail.ncku.edu.tw; 13Institute of Behavioral Medicine, College of Medicine, National Cheng Kung University, Tainan 701, Taiwan; 14Department of Psychiatry, Tainan Hospital, Ministry of Health and Welfare, Tainan 700, Taiwan; 15Department of Psychiatry, Kaohsiung Medical University Hospital, Kaohsiung 807, Taiwan; sheng@kmu.edu.tw; 16School of Medicine, Kaohsiung Medical University, Kaohsiung 807, Taiwan; 17Institute of Brain Science, Brain Research Center, National Yang Ming Chiao Tung University, Taipei 112, Taiwan; ctli2@vghtpe.gov.tw; 18Division of Psychiatry, Faculty of Medicine, National Yang Ming Chiao Tung University, Taipei 112, Taiwan; 19Mind-Body Interface Laboratory (MBI-Lab), Department of Psychiatry, China Medical University Hospital, Taichung 404, Taiwan; cobolsu@gmail.com; 20College of Medicine, China Medical University, Taichung 404, Taiwan; 21An-Nan Hospital, China Medical University, Tainan 709, Taiwan; 22Institute of Medicine, Chung Shan Medical University, Taichung 402, Taiwan; 55810@cch.org.tw; 23Department of Psychiatry, Chung Shan Medical University Hospital, Taichung 402, Taiwan; 24Department of Psychiatry, Chang Gung Memorial Hospital at Linko, Taoyuan 333, Taiwan; 25College of Medicine, Chang Gung University, Taoyuan 333, Taiwan

**Keywords:** cognitive impairment, major depressive disorder, Taiwan Cognition Questionnaire (TCQ), Taiwanese Depression Questionnaire (TDQ)

## Abstract

Cognitive dysfunction is associated with functional impairment of patients with Major Depressive Disorder (MDD). The goals were to explore the associated factors of cognitive impairment in MDD and to develop and validate a brief and culture-relevant questionnaire, the Taiwan Cognition Questionnaire (TCQ), among patients with MDD. This was a cross-sectional, multi-center observational study of MDD patients in Taiwan. Participants of Group 1 from 10 centers contributed to the validation of the TCQ by their response and sociodemographics. The participants of Group 2 from one center received an objective cognitive assessment for clarification of the relationship between the TCQ score and its associated factors. In Group 1, 493 participants were recruited. As for Group 2, an extra 100 participants were recruited. The global Cronbach’s alpha for the TCQ was 0.908. According to the coordinates of the ROC curve, 9/10 was the ideal cut-off point. With the criteria, the sensitivity/specificity of the TCQ was 0.610/0.689. The TCQ score was positively associated with a history of being admitted to acute psychiatric care and the severity of depression and negatively associated with objective cognitive measures. The TCQ provides a reliable, valid, and convenient measure of subjective cognitive dysfunction in patients with MDD.

## 1. Introduction

Major Depression Disorder (MDD) is a common and disabling mental disorder around the world. MDD can cause physical and psychological dysfunction and lead to premature death [1]. Early in the last century, the World Health Organization postulated that MDD would be one of the two top causes of disease burden measured by Disability-Adjusted Life Years, second only to ischemic heart disease. In 2010, MDD was the second leading medical cause of burden globally, with the highest estimates of disability in people of working age [2]. The consequences will be even worse after the COVID-19 pandemic [3].

Many studies have shown that cognitive dysfunction is a powerful predictor of occupational and social functional impairment in adults with MDD. The persistence of cognitive deficits after remission of depressive symptoms has been shown to contribute to the failure in achieving full functional recovery in MDD [4].

Domain-specific mild to moderate cognitive deficits in learning and memory, executive function, processing speed, and attention and concentration are highly replicated findings in individuals with MDD [5]. Cognitive impairment, as an epiphenomenon of MDD, is a core domain disturbance of MDD and associated with, but independent of, mood symptoms [6]. The persistence of cognitive impairment in remitted phases of MDD is related to psychosocial dysfunction in terms of quality of life, social, occupational, and global functioning [7]. Recent research also reveals that objective cognitive dysfunction negatively affects the organization and occupational and social functioning and may lead to poor treatment response and recurrence of MDD [8]. Early identification, intervention, and follow-up for cognitive impairment in patients with MDD are crucial in designing a comprehensive treatment plan. To achieve this goal, standardized cognitive screening tools are key to the identification and assessment of treatment outcomes for cognitive dysfunction in MDD [9].

Subjective cognition is easily influenced by mood state and may well be affected by the severity of depressive symptoms to a greater extent than objective cognition. Although subjective cognition is informative of perceived dysfunction, it may not exactly reflect objective cognitive ability [10]. Objective measures, such as performance-based neuropsychological tests, are precise and effective in assessing the cognitive function of patients with MDD. While clinicians strive to screen cognitive dysfunction amongst MDD patients, screening instruments should be brief, easy to administer, and capable of detecting objective cognitive impairment [11]. Several standard screening tools have been developed for identifying cognitive dysfunction in MDD. Some of them are objective, such as the Screen for Cognitive Impairments in Psychiatry-D (SCIP-D) [12], and have to be performed by trained staff instead of MDD patients themselves. Others are subjective, such as the Cognitive Complaints in Bipolar Disorder Rating Assessment (COBRA) [13] and the Perceived Deficits Questionnaire (PDQ-20) [14], yet the wording of cognitive symptoms of those questionnaires may not be easily understood by MDD patients of different cultural backgrounds or regions. Some measures incorporate both objective and subjective tools, such as the THINC-Integrated Tool (THINC-it) for assessing cognitive dysfunction in MDD [10], requiring more time and equipment to complete the assessment.

The primary aim of this study was an exploration of the prevalence of cognitive impairment among MDD patients and its associated factors in a national representative sample in Taiwan. The second was the development and validation of a brief and culture-relevant questionnaire, the Taiwan Cognition Questionnaire (TCQ), which could be used as a domain-specific and objective measure of cognitive function in MDD.

## 2. Materials and Methods

### 2.1. Subjects

There were two groups of participants in this study. Group 1 participants were recruited from 10 centers in all regions of Taiwan. The size of the sample for each region was approximately proportionate to the population distribution. Participants needed to be 18 or older, capable of communicating in Chinese, and to have consented to participate in the study. The psychiatric diagnosis (ICD-10) of the participants selected was F32.x or F33.xx, major depressive disorder, single episode or recurrent. Of the selected participants, no one had ever been diagnosed with other diseases that might have had an impact on their cognitive function, such as dementia, organic mental disorder, substance use disorder, or could not comply with all the conditions of the study. In Group 2, one of the 10 centers (the E-Da Hospital) recruited sufficient extra participants with the same criteria and arranged further assessment of cognitive function and depressive symptoms.

### 2.2. Study Design

This was a cross-sectional study with multiple centers. The study was reviewed and approved by the Institutional Review Board of the E-da Hospital. The participants provided their written informed consent to participate in this study. In Group 1, all centers contributed to the recruitment of the representative sample of MDD in Taiwan in validating the TCQ. Participants’ socio-demographic data, clinical characteristics, and responses to TCQ were collected. In Group 2, participants were assessed with objective measures to define the suitable cut-off points of TCQ for identifying cognitive dysfunction in MDD. For exploring the relationship between TCQ score and its associated factors, depression severity was evaluated by using the Taiwanese Depression Questionnaire (TDQ) [15], a convenient subjective measure for depressive symptoms, which includes four objective cognitive measures described.

### 2.3. Subjective Measure

The TCQ used had 5 questions with a 4-point ordinal categorical response scale to reflect the frequency of experiencing a specific cognitive problem over the past 7 days (Appendix A). Scores for each of the 5 questions were calculated by assigning a value of 0 (“never or rarely—never or once in the past 7 days”), 1 (“sometimes—twice or thrice”), 2 (“often—4–5 times”), or 3 (“very often—6–7 times”) to each item. The five items were summed to produce a score ranging from 0 to 15. The higher the total score, the greater the perceived cognitive dysfunction. These five questions were based on clinical experience in treating patients with MDD, and appropriate wording was used with the aim of noting common cognitive complaints in terms of memory, executive function, speed of processing, attention, and decision making. The questionnaire could be completed either by patients themselves or trained interviewers. An English translation of the TCQ is listed in Table 1. A pre-test was performed before the study was conducted and before TCQ was completed by all participants in Groups 1 and 2. For statistical analysis, the total score was treated as a continuous variable representing the severity of cognitive dysfunction in MDD. The higher the score, the worse the cognition impairment.

### 2.4. Objective Measures

The following four neuropsychological tests have been widely used and well established in assessing the cognitive impairment in MDD patients, and all the Group 2 participants underwent a neuropsychological test by a single experienced clinical psychologist.

#### 2.4.1. Digit Symbol Substitution Test (DSST)

DSST is related to the speed of response, attention, and executive function in MDD patients. DSST is also sensitive to the change of cognitive function of MDD patients in their treatment course. In this study, we applied the Wechsler Adult Intelligence Scale-Forth Edition—Coding (WAIS-IV—Coding) to perform the DSST. The Wechsler Adult Intelligence Scale-Forth Edition, Taiwan version (WAIS-IV) is an adult intelligence test consisting of ten core subtests and five supplementary subtests. The Coding subtest was used to measure processing speed [16]. In the test, one point is given for each correctly drawn symbol (completed within 120 s), and the total raw score is the sum of each point. The raw score was converted into scaled scores (mean = 10, SD = 3) by reference to the norm. We used the percentile scale of the scaled score conversion as an indicator to represent the participants’ processing speed and executive function.

#### 2.4.2. Digit Span Forwards and Backwards (DS)

DS is thought to be correlated with attention, processing speed, and working memory in MDD patients. DS is also closely related to MDD patients’ cognitive deficit. In this study, we applied the Wechsler Memory Scale-Third Edition—Digit Span (WMS-III—Digit Span) to perform the DS. The Digit Span subtest of the WMS-III was used to measure attention and working memory [17]. The subtest consisted of a forward and a backward element. The forward span has been argued to represent a measure of the capacity of the attention span. In the forward span task, the experimenter read out a series of numbers and participants were asked to recite the numbers. The difficulty of the task was increased as the number of digits increased. This measures the amount of information participants can store. The second part is the backward span, measuring the attention span and the working memory. The experimenter again read out a series of numbers, but participants were asked to repeat them in reverse order. The participants had to rearrange the number and process the information before it could be repeated. The total score was the sum of the forward and backward-span tasks converted into a scaled score (mean = 10, SD = 3) by reference to the norm. This study used the percentile scale of the scaled score conversion as an indicator to represent the participants’ performance in attention and working memory.

#### 2.4.3. Verbal Fluency Task (VFT)

VFT is supposed to be related to the ability to retrieve information from memory and is, therefore, related to a cognitive deficit in MDD. Semantic association of the verbal fluency test involves the integration of abilities such as information processing speed and semantic knowledge. It also relies on the strategic development of executive function and self-monitoring [18]. Participants were asked to name three categories of objects, fruits, fish, and vegetables, in sequence. Each category was given 60 s, and the total score was the sum of the number of objects that they correctly named in each category. In this study, we compared the norm age and education level in Taiwan to obtain the percentile scale as a statistical analysis indicator.

#### 2.4.4. Wechsler Memory Scale-Third Edition—Word Lists (WMS-III—Word Lists, WL)

The Wechsler Memory Scale-Third Edition, Taiwan version (WMS-III) is a set of tools for measuring language, memory, and working memory, mainly measuring the declarative part of episodic memory. The testing materials are novel and specific, and participants are required to comprehend text and context and thus learn and acquire new information [17]. The Word List subtest was used to evaluate the ability of verbal learning and memory. During the test, participants were asked to recall 12 pairs of unrelated words presented orally in four rounds. The total score was the sum of the four rounds of scores converted into a scaled score (mean = 10, SD = 3) by reference to the norm. This study used the percentile scale of the scaled score conversion as an indicator of the verbal learning and memory ability of the participant.

### 2.5. Statistical Analyses

#### 2.5.1. Descriptive Statistics

Socio-demographics, clinical characteristics, subjective cognitive measure (TCQ), and objective cognitive measures were analyzed by calculating their means with standard deviation or counts with the percentage in the two groups.

#### 2.5.2. Validity and Reliability Testing of TCQ

For content validity, specific validation of TCQ content was performed based on expert review. For construct validity in Group 1, the determination of the adequacy of the exploratory factor analysis (EFA) was performed using Bartlett’s test and the Kaiser–Meyer–Olkin (KMO) measure. The 5 qualitative items of the questionnaire were then submitted to the EFA. Principal factor analysis with a varimax rotation was used to explore the structure underlying the 5 qualitative items. The inclusion or exclusion of an item in a construct was determined iteratively by examining factor loadings and Cronbach’s alpha to identify redundant items or those that did not sufficiently measure the same underlying construct. The reliability of TCQ was further measured based on its internal consistency using Cronbach’s alpha and the split-half reliability method.

#### 2.5.3. Cut-Off Points of TCQ in Identifying Cognitive Impairment

Receiver’s Operating Curve (ROC) analysis was used on the TCQ scores of Group 2 participants to determine their subjective cognitive performance and to find a reasonable cut-off point for predicting cognitive dysfunction by TCQ score. An outcome of 10 percentile or less in the DSST, DS, VFT and WL tests was defined as cognitive impairment.

#### 2.5.4. TCQ Scores Predicted by Associated Factors

Hierarchical multivariate linear regression was applied to analyze TCQ score, and associated factors among participants from Group 2. Personal characteristics (age, sex, educational level, admission to an acute psychiatric care unit), cognitive composite score (results of the 4 neuropsychological tests in percentiles), and depression severity (measured by TDQ) were included in the analysis.

## 3. Results

### 3.1. Descriptive Statistics

Between December 2020 and February 2021, 593 MDD patients were recruited for this study in 10 different centers in Taiwan after approval by the IRB. The numbers, geographical regions, socio-demographics, and clinical characteristics are summarized in Table 2. In Group 1, the sample was representative in terms of their distribution and proportion of the general population. A total of 190 participants were recruited from 4 centers in northern Taiwan; 153 participants from 3 centers in middle Taiwan; and 150 participants from 3 centers in southern Taiwan. The mean age was 49.9 years (SD 17.1); the mean age of onset of MDD was 39.5 years (SD 16.2); the mean duration since onset was 10.3 years (SD 9.2). Most participants were female (67.7%), married (51.7%), had college or higher education (40.6%), and unemployed (70.8%). Only 32% of the participants had been admitted to the acute psychiatric care unit. More participants were diagnosed as F33.2 (Major depressive disorder, recurrent, severe without psychotic features) and F33.1 (Major depressive disorder, recurrent, moderate) (21.3% and 13.8%, respectively) when recruited. The mean TCQ score was 7.1 (SD 4.8). Mean scores across all TCQ items were generally in the range of 1.25 to 1.54, and SDs were generally between 1.09 and 1.14 (Table 3). Floor and ceiling effects were minimal for the TCQ total score, with values generally below 10%.

The mean age of the extra participants that had been recruited for Group 2, was similar (52.2 years (SD 12.10)), mean age at onset (41.9 years (SD 13.1)), mean duration since onset (10.4 years (SD 8.20)), and a similar proportion in sex (female 67.3%), marital status (married 48.7%), and employment (unemployed 70.7%). A comparison with participants recruited from all centers showed those from the E-Da hospital had fewer with college or higher education (20.0%), and they also had a higher chance of being admitted to an acute psychiatric care unit (44%). More participants in the E-Da Hospital group were diagnosed as F33.2 (Major depressive disorder, recurrent, severe without psychotic features) and F33.3 (Major depressive disorder, recurrent, severe with psychotic features) (33.3%, 13.3%, respectively) when recruited. Their mean TCQ score was 9.36 (SD 4.51); the mean TDQ score was 31.67 (SD 14.32). The socio-demographics, clinical characteristics, subjective measures for depression and cognitive, and objective cognitive measures of 150 participants are listed in Table 2.

### 3.2. Validity and Reliability Testing of TCQ

#### 3.2.1. Content Validity

Ten experts from different geographic locations of Taiwan evaluated the questionnaires. Revisions were made after thorough discussions, and consensus was reached without any objections.

#### 3.2.2. Construct Validity

Based on the response of TCQ from participants of Group 1, the Kaiser–Meyer–Olkin measure of sampling adequacy was 0.88. The significance for Bartlett’s test of sphericity was less than 0.001. The screen plot showed that only one component was extracted.

#### 3.2.3. Reliability

All of the five questions of TCQ were included, and the global Cronbach’s alpha for the questionnaires was 0.908 for the five items in the completed TCQs for Group 1.

### 3.3. Cut-Off Points of TCQ in Identifying Cognitive Impairment

The area under the ROC curve was 0.685 (95% CI 0.595 to 0.776, Asymptotic sig < 0.001, see Figure 1). According to the coordinates of the ROC curve (Table 4), 9/10 was the appropriate cut-off point for defining possible cognitive impairment in Group 1 participants. With these criteria, the sensitivity/specificity of TCQ was 0.610/0.689 in identifying cognitive impairment. In Group 2, 105 (70.0%) out of 150 MDD patients might have been cognitively impaired. When the same criteria were applied to Group 1, 199 (33.5%) out of 493 MDD patients might have been cognitively impaired. Furthermore, in Group 1, with a prevalence of cognitive dysfunction at 33.5%, the positive predictive value (PPV) of TCQ was 0.497, and the negative predictive value (NPV) was 0.778. In Group 2, with a prevalence of cognitive dysfunction at 70.0%, the PPV/NPV of TCQ was 0.821/0.431.

### 3.4. TCQ Scores Predicted by Associated Factors

From the outcomes of hierarchical multivariate linear regression (see Table 5), three models were formulated with the R squares ranging from 0.038 to 0.684. In Model 1, none of the socio-demographic factors were associated with the TCQ score. In Model 2, ever having been admitted to the acute psychiatric care unit in the past was positively associated with the TCQ score. This was after control for all socio-demographic factors, and objective cognitive function (beta coefficient 2.417, 95% CI 0.942 to 3.892) had been applied and clearly shows that previous illness severity was significantly associated with more impairment in subjective cognitive function as measured by TCQ. While composite cognitive score was negatively associated with TCQ score, after control for all socio-demographic factors and previous illness severity (beta coefficient −0.014, 95% CI −0.022 to −0.005), it can be clearly seen that a better objective cognitive function was significantly associated with a better subjective cognitive function as measured by TCQ.

In Model 3, ever having been admitted to the acute psychiatric care unit in the past was still positively associated with TCQ score but attenuated when controlled for all socio-demographic factors, objective cognitive function, and present depression severity (beta coefficient 1.194, 95% CI 0.264 to 2.123). Again, this implied that more previous illness severity was significantly associated with more impairment in subjective cognitive function measured by TCQ. While composite cognitive score was not significantly associated with TCQ score when controlled for all socio-demographic factors, previous illness severity, and depression severity (beta coefficient −0.003, 95% CI −0.008 to 0.003), depression severity measured by TDQ score was associated positively with TCQ score when controlled for all socio-demographic factors, previous illness severity, and objective cognitive function (beta coefficient 0.248, 95% CI 0.215 to 0.280). Clearly, severe existing depression was significantly associated with more impairment in subjective cognitive function measured by TCQ.

## 4. Discussion

The TCQ provides a convenient measure of subjective cognitive dysfunction in patients with MDD with fair sensitivity/specificity (0.610/0.689) when the cut-off point is 9/10. Participants’ depressive symptom severity and the history of ever having been admitted to an acute psychiatric care unit are significant positive TCQ score predictors. In this study, the prevalence of cognitive impairment of the representative sample recruited from 10 centers (Group 1) in Taiwan was 40.4%; the prevalence of cognitive impairment from Group 2 was 70.0%. With the low sensitivity/specificity in this study, the TCQ is considered a screening tool rather than a diagnostic instrument for identifying possible cognitive dysfunction before referral for detailed assessment of cognitive function in patients with MDD. The objective cognitive measure, the composite cognitive core, is also a significant predictor of the TCQ score. Several different study methods have shown that cognitive impairment affects approximately one-third to one-half of previously diagnosed MDD patients [19]. The prevalence of cognitive impairment in MDD patients of our national representative sample is consistent with previous research. However, the prevalence in Group 2 was higher. It is important to note that there are several factors, such as age [20], age at onset of MDD [21], level of education [22], MDD subtype [23], depression severity [24], treatment regimen [25], duration of untreated MDD [24], childhood adversity [26], and physical activity [21], that have been demonstrated to influence cognitive performance in patients with MDD. The discrepancy of prevalence for cognitive impairment between the two groups in our study may be partly due to the difference in educational level, MDD subtype, and depression severity. On the other hand, pharmacotherapy regimens have not been included or analyzed in our study. Current pharmacotherapeutic options in ameliorating cognitive dysfunction of patients with MDD or schizophrenia remain limited and poorly studied [27]. There is recent evidence that supports serotonin-norepinephrine reuptake inhibitors rather than selective serotonin reuptake inhibitors for improving verbal and visual memory [28], as well as vortioxetine for numerous domains of cognition [29]. In our study, all patients with MDD received their normal pharmacological treatment without interruption. The lack of information of antidepressants and adherence is a limitation for analysis or adjustments for the effect of treatment. In addition to the severity of illness, its duration, untreated duration, and treatment response, the presence of medical co-morbidities may further impact cognitive function. For example, metabolic co-morbidities, such as obesity, are associated with cognitive dysfunction and are often seen in depressed people [30]. The lack of information about medical co-morbidities in participants may be another limitation of this study. However, the TCQ provides a good start in the quick identification of cognitive impairment in MDD despite the moderate sensitivity and specificity and limitations, such as small sample size and the lack of thorough information of treatment and adherence in this study.

Cognitive impairment in MDD patients is so prevalent that clinicians need to take cognitive dysfunction and its impact into consideration while making a comprehensive treatment plan. The personal daily function of a depressed person is closely related to their depression and directly mediated by cognitive function [31]. The pathophysiology for cognitive dysfunction in MDD may involve multiple biological pathways, including hypothalamic-pituitary-adrenal axis hyperfunction, more oxidative and nitrosative stress, inflammation, mitochondrial dysfunction, increased apoptosis, and diminished neurotrophic support [32]. These heterogenous etiologies suggest the identity of several promising neurotherapeutic targets, such as minocycline, statins, anti-inflammatory compounds, N-acetylcysteine, omega-3 polyunsaturated fatty acids, erythropoietin, thiazolidinediones, glucagon-like peptide-1 analogs, S-adenosyl-l-methionine, cocoa flavanols, creatine monohydrate, and lithium [32]. In addition, some non-pharmacological strategies, such as cognitive remediation and exercise, are being studied in patients with MDD though the study quality may not be quite satisfactory [11]. Cognitive remediation is also effective in alleviating neurocognitive symptoms in patients with MDD [33]. Cognitive behavior therapy and memory-specific therapies may reduce cognitive vulnerability, prevent depressive relapse, and improve subjective complaints by modifying negative memory bias in patients with MDD [34]. More recently, repetitive transcranial magnetic stimulation (rTMS) has emerged as a neurostimulation method for diminishing the propensity for cognitive impairment in patients with MDD [35]. rTMS is less invasive in nature and more favored than electroconvulsive therapy. A systematic review exploring the role of therapeutic rTMS course reveals that administration of rTMS to the prefrontal cortex for depression may produce modest cognitive-enhancing effects in psychomotor speed, visual scanning, and set-shifting ability [36]. However, novel interventions for improving cognitive function in patients with MDD are still needed as present interventions for cognitive impairment in MDD do not meet the criteria for functional recovery [32,37].

More vigorous studies with larger sample sizes, longer follow-up periods, and more various interventions are expected. To attain this goal, a standardized and convenient tool for assessing cognitive function in MDD is necessary. TCQ is a brief and effective tool for assessing cognitive impairment in MDD patients in daily psychiatric practice in Taiwan with good validity and reliability. TCQ is applicable to all adult MDD patients with various depression severity in different clinical situations, including outpatient, inpatient, and day care services. The detection of cognitive impairment in MDD by using TCQ may help clinicians manage this issue properly and efficiently. TCQ is also convenient in the evaluation of cognitive function improvement for MDD patients receiving treatment. This is even more important when we conduct research aiming to develop new interventions for cognitive dysfunction of MDD in the future. Compared to other subjective screening measures for cognitive impairment in MDD, such as COBRA [9,13], TCQ is faster and more user-friendly (5 questions vs. 16 questions in COBRA) and has similar sensitivity (0.61 vs. 0.65) and specificity (0.69 vs. 0.68). TCQ is further correlated with objective cognitive measures while COBRA is not. When it comes to PDQ-D [14], a 20-item, reliable, and valid measure of subjective cognitive dysfunction in patients with MDD in the UK and US, TCQ is concise and convenient for Chinese-speaking patients with MDD. For other objective cognitive measures, such as SCIP-D [12], or composite cognitive measures, such as THINC-it [10], trained staff, specific device, and more time is needed compared to TCQ.

## Figures and Tables

**Figure 1 jpm-12-00359-f001:**
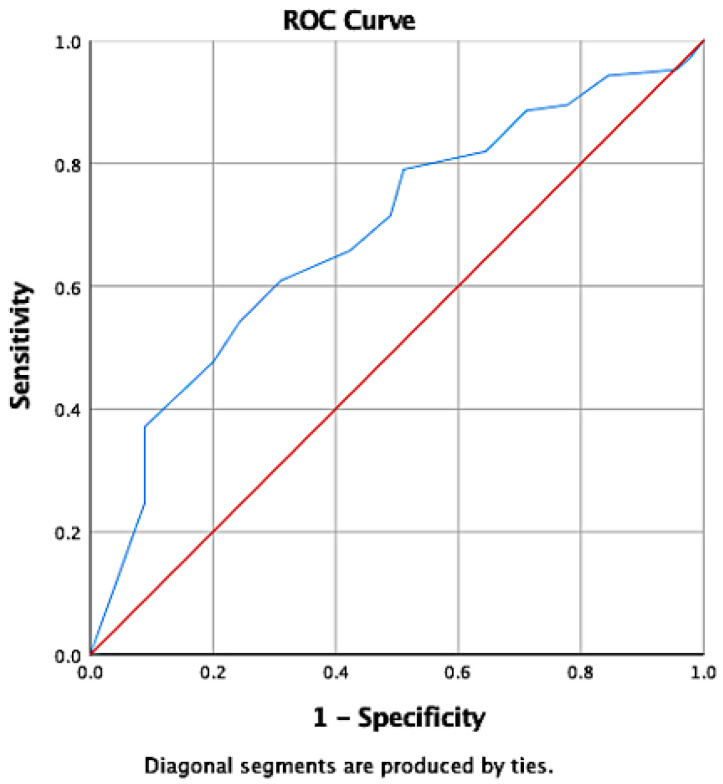
ROCs of TCQ scores determined by cognitive dysfunction in Group 2.

**Table 1 jpm-12-00359-t001:** Taiwan Cognition Questionnaire (TCQ).

During the Past 7 Days, How Often Was…	Never or Rarely (0 or Once)	Sometimes (Twice or Thrice)	Often (4 to 5 Times)	Very Often (6 to 7 Times)
My memory worse than usual	0	1	2	3
My attention worse than usual	0	1	2	3
I slower than usual	0	1	2	3
It more difficult for me to make a decision than usual	0	1	2	3
I less capable than usual	0	1	2	3

**Table 2 jpm-12-00359-t002:** Socio-demographics and characteristics of participants by group.

		Group 1 (*N* = 493)		Group 2 (*N* = 150)	
		Mean/Count	SD/%	Mean/Count	SD/%
Age (years)		49.9	17.1	52.2	12.1
Age of onset (years)		39.5	16.2	41.9	13.1
Years since onset		10.3	9.2	10.4	8.2
Sex	male	159	32.3	49	32.7
	female	334	67.7	101	67.3
Marital status	married	255	51.7	73	48.7
	single	145	29.4	26	17.3
	divorced/widowed	93	18.9	51	34.0
Education	primary or less	72	14.6	26	17.3
	middle	62	12.6	38	25.3
	high	159	32.3	56	37.3
	college or more	200	40.6	30	20.0
Occupation	unemployed	349	70.8	106	70.7
	full-time	114	23.1	17	11.3
	part-time	30	6.1	27	28.0
Ever hospitalized ^1^	yes	158	32.0	66	44.0
	no	335	68.0	84	56.0
ICD-10 diagnosis	F32.0	19	3.9	1	0.7
	F32.1	27	5.5	7	4.7
	F32.2	13	2.6	12	8.0
	F32.3	5	1.0	1	0.7
	F32.4	11	2.2	0	0
	F32.5	15	3.0	0	0
	F32.9	24	4.9	9	6.0
	F33.0	22	4.5	1	0.7
	F33.1	68	13.8	16	10.7
	F33.2	105	21.3	50	33.3
	F33.3	20	4.1	20	13.3
	F33.40	38	7.7	2	1.3
	F33.41	55	11.2	1	0.7
	F33.42	19	3.9	0	0
	F33.9	52	10.5	30	20.0
TCQ ^2^ score		7.10	4.80	9.36	4.51
TDQ ^3^ score		NA	NA	31.67	14.32
DSST ^4^ percentile		NA	NA	22.5	25.5
DS ^5^ percentile		NA	NA	30.7	27.5
VFT ^6^ percentile		NA	NA	36.4	31.8
WL ^7^ percentile		NA	NA	30.0	26.3

^1^ Ever hospitalized: ever admitted to the acute psychiatric care unit; ^2^ TCQ: Taiwan Cognition Questionnaire; ^3^ TDQ: Taiwanese Depression Questionnaire; ^4^ DSST: Digit Symbol Substitution Test; ^5^ DS: Digit Span Forwards and Backwards; ^6^ VFT: Verbal Fluency Task; ^7^ WL: Wechsler Memory Scale-Third Edition—Word Lists.

**Table 3 jpm-12-00359-t003:** Item statistics and internal consistency reliability of TCQ from Group 1.

Item No.	Mean	SD	Floor (%)	Ceiling (%)	Cronbach’s Alpha	Item Total Correlation
1	1.54	1.09	20.1	26.6	0.907	0.673
2	1.39	1.14	29.0	23.5	0.873	0.853
3	1.48	1.10	23.9	24.3	0.881	0.802
4	1.43	1.14	27.8	24.9	0.884	0.787
5	1.25	1.14	35.5	19.9	0.890	0.746
total	7.10	4.80	8.50	9.50	--	--

**Table 4 jpm-12-00359-t004:** Coordinates of the Curve of TCQ * total score with cognitive dysfunction in Group 2.

Cut-Off Points	Sensitivity	1-Specificity	Youden’s Index
2/3	0.943	0.844	0.099
3/4	0.895	0.778	0.117
4/5	0.886	0.711	0.175
5/6	0.819	0.644	0.175
6/7	0.790	0.511	0.279
7/8	0.714	0.489	0.225
8/9	0.657	0.422	0.235
9/10	0.610	0.311	0.299
10/11	0.543	0.244	0.299
11/12	0.476	0.200	0.276
12/13	0.371	0.089	0.282
13/14	0.343	0.089	0.254
14/15	0.248	0.089	0.159

* TCQ: Taiwan Cognition Questionnaire.

**Table 5 jpm-12-00359-t005:** Hierarchical multivariate linear regression predicting TCQ ^1^ scores in Group 2.

Model	Variable	Unstandardized		Standardized Beta	95% CI for B	*p*	R^2^
		B	SE				
1	Age (years)	−1.012	0.839	−0.106	−2.670 0.645	0.229	0.038
	Male (vs. female)	−0.027	0.034	−0.073	−0.094 0.039	0.419	
	Married (vs. not-married)	−0.807	0.770	−0.090	−2.329 0.715	0.296	
	Education < 9 years (vs. >9 years)	0.248	0.798	0.027	−1.330 1.825	0.757	
	Unemployed (vs. else)	0.561	0.866	0.057	−1.151 2.273	0.518	
2	Age (years)	−1.234	0.785	−0.129	−2.785 0.317	0.118	0.175
	Male (vs. female)	0.007	0.033	0.018	−0.058 0.071	0.842	
	Married (vs. not-married)	−0.195	0.749	−0.022	−1.676 1.286	0.795	
	Education < 9 years (vs. >9 years)	−0.499	0.766	−0.055	−2.013 1.015	0.516	
	Unemployed (vs. else)	−0.321	0.829	−0.032	−1.959 1.318	0.699	
	Ever hospitalized ^2^ (vs. never)	2.417	0.746	0.267	0.942 3.892	0.001	
	Cognition composite score ^3^	−0.014	0.004	−0.260	−0.022 −0.005	0.002	
3	Age (years)	−0.110	0.493	−0.011	−1.084 0.864	0.824	0.684
	Male (vs. female)	0.013	0.020	0.034	−0.027 0.053	0.533	
	Married (vs. not-married)	0.728	0.469	0.081	−0.199 1.656	0.123	
	Education < 9 years (vs. >9 years)	0.208	0.478	0.023	−0.737 1.152	0.665	
	Unemployed (vs. else)	−0.067	0.515	−0.007	−1.084 0.951	0.897	
	Ever hospitalized (vs. never)	1.194	0.470	0.132	0.264 2.123	0.012	
	Cognition composite score ^3^	−0.003	0.003	−0.055	−0.008 0.003	0.294	
	TDQ ^4^ total score	0.248	0.016	0.786	0.215 0.280	<0.001	

^1^ TCQ: Taiwan Cognition Questionnaire; ^2^ Ever hospitalized: ever admitted to the acute psychiatric care unit; ^3^ Cognition composite score: sum of four percentiles of neuropsychological tests; ^4^ TDQ: Taiwanese Depression Questionnaire.

## Data Availability

Data are open to the public.

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
