# Peer review of "A Multi-Center Study for the Development of the Taiwan Cognition Questionnaire (TCQ) in Major Depressive Disorder"

_jpm, 2022, doi:10.3390/jpm12030359_

Round 1
Reviewer 1 Report
Yen and colleagues present an interesting work regarding a neglected dimension of mood disorders, the impaired cognition at both subjective and objective levels. They developed what the called the Taiwan Cognition questionnaire (TCQ) to capture subjective cognitive dysfunction in two groups (multicentre and single centre oriented) groups of major depressive disorders (MDD) patients in Taiwan. The methodology is impeccable.
My major concern is the “optimistic” predictive performance, given the modest discriminatory performance of TCQ. The specificity and selectivity are below the gold standard 0.8 cut-off (Polo TCF, Miot HA. Use of ROC curves in clinical and experimental studies. J Vasc Bras 2020,19:e20200186). This does not by any way demerit the scientific soundness of the work. Indeed, the AUC of the ROC is not the only measure of test validity. The variables under scrutiny are not as tangible as the levels of a specific metabolite in blood as a biomarker of certain types of cáncer, to say so. To get my point, do not mix prognosis with discriminative power (the ROC approach), because these measurable variables pertain to the realms of Psychiatry, and as such, their boudaries are very unprecise.
In my opinion, the authors should state more clearly in the discussion that this TCQ approach is a good start despite the limitations (e.g., small size of the simple and the lack of information about the adherence to treatments to treat mood and affect disorders, etc.).
My second major concern is that the first objective of the study was to demonstrate the nexus between mood disorders and cognitive dysfunction. This is a very great research lead. However, this is not conveniently stated in the conclusion paragraph. The authors should make a point out of it. Please, use the first paragraph to clearly define the main findings of your research that answer the two objectives put forward at the end of the discussion. Do not spend too much time explaining the effects of the medication patients are supposed to take to treat their depression. You do not have this piece of information.
My last concern is the lack of references regarding the DSST, DS, and WMS-III methods (there is already one for the VFT). Please, include a very brief statement of ethics in the Methods section, and not only at the end of the manuscript.
Minor concerns:
What does the acronym THINC stand for?
Line 1109: “of the selected participants”
Lines 132-113 are similar to lines 140-141.
Line 201 “is the two groups”.
Author Response
Please see attached response. Thanks!

Reviewer 2 Report
Authors try to stablish a nationalize Questionnaire to detect cognitive deficiency. The topic is interesting and study enrolled relatively a large sample size. It seems novel among Tawnies population.
Abstract
Background: please add some sentences about the importance of study subject.
Method: this part is very messed up. Briefly explain type of study, study participant, place of study and etc.
Conclusion: please delete this sentence: Cognitive impairment 49 in MDD involves present and severity of past illnesses.
Introduction:
- Is too long. It is very difficult to follow the main subject in this part.
- Please rewrite in approximately 500 words and highlight why using of a nationalize MDD questionnaire could be useful to detect patients and set up their treatment.
- Authors should highlight the necessity of this study and explain why standard questionnaire can not be useful for determining MDD in Taiwanese population.
Method:
- Study design: Present key elements of study design early in the paper
- Setting: Describe the setting, locations, and relevant dates, including periods of recruitment, exposure, follow-up, and data collection (a) Give the eligibility criteria, and the sources and methods of case ascertainment and control selection. Give the rationale for the choice of cases and controls.
- Participants: (b) For matched studies, give matching criteria and the number of controls per case
- Variables: Clearly define all outcomes, exposures, predictors, potential confounders, and effect modifiers. Give diagnostic criteria, if applicable.
- Data sources/ measurement 8* For each variable of interest, give sources of data and details of methods of assessment (measurement). Describe comparability of assessment methods if there is more than one group.
- Bias: Describe any efforts to address potential sources of bias.
- Describe any methods used to examine subgroups and interactions (c) Explain how missing data were addressed (d) If applicable, explain how matching of cases and controls was addressed.
- Statistical methods: Describe any sensitivity analyses.
Results: Indicate number of participants with missing data for each variable of interest
Discussion:
Summaries key results with reference to study objectives
Discuss limitations of the study, taking into account sources of potential bias or imprecision. Discuss both direction and magnitude of any potential bias
Give a cautious overall interpretation of results considering objectives, limitations, multiplicity of analyses, results from similar studies, and other relevant evidence
Discuss the generalizability (external validity) of the study results
Author Response
Please see attached response. Thanks!

Reviewer 3 Report
Overall, the manuscript lacks a clear focus and also requires a close edit for language and grammar, ideally by a native English speaker.
Specific comments:
- The study title itself does not flow well. I am not sure how "Cognitive Impairment in Major Depressive Disorder" leads to "A Multi-Center Study for the Development of the Taiwan Cognition Questionnaire (TCQ)". The first part of the title could be omitted.
- The abstract should be a total of about 200 words maximum. The abstract should be a single paragraph and should follow the style of structured abstracts, but without headings.
- "In Group 2, one center recruited extra sufficient participants and arranged objective assessment and exploring the relationship between the TCQ score and its associated factors" - very awkward phrasing, please rephrase.
- "Cognitive impairment in MDD involves present and severity of past illnesses" - do you mean involves both present and past severity of illness?
- The first paragraph is too long and convoluted. It could be better divided into two separate paragraphs.
- Cognitive dysfunction in major depression encompasses several domains, most commonly small to moderate deficits in attention, memory, learning, processing speed and executive function (citation: ncbi.nlm.nih.gov/pubmed/26801406). Importantly, it is often an epiphenomenon, correlating with, but dissociable from, affective symptoms; impairments in cognitive functioning (as well as in psychosocial functioning) may persist even after patients have met conventional criteria for remission of depressive symptoms (citation: ncbi.nlm.nih.gov/pubmed/26290264).
- It is relevant to mention that current pharmacotherapy options to treat cognitive dysfunction in depressed patients or patients with schizophrenia remain extremely limited and poorly studied (citation: pubmed.ncbi.nlm.nih.gov/31742775).
- "... culture-relevant expressions in cognitive complaints" - but the present study and interviews were only conducted in Mandarin, is that correct?
- On average, how long did it take for participants to complete the questionnaire?
- The sample size is actually quite small and mostly female, meaning that it is difficult to make a generalization to the broader population.
- The flow from "Group 1" to "Group 2" is unclear. Perhaps would be better to call these two different groups more directly, what was the main purpose for Group 1 vs Group 2?
- The underlying data should be made publicly available. If this was not possible, please provide a reason why.
- A copy of the TCQ should be appended for review.
Author Response
Please see attached response. Thanks!

Round 2
Reviewer 1 Report
None
Author Response
Thank you for your review and advice.
Reviewer 2 Report
Accept.
Author Response
Thank you for your review and advice.
Reviewer 3 Report
Thank you for the revisions.
Specific comments:
- "Participatants" is misspelled.
- "According to the coordinates of the ROC curve, 9/10 was the ideal cut-off point. With the criteria, the sensitivity/specificity of the TCQ was 0.610/0.689" - this is actually clinically not useful given the relatively low sensitivity and specificity even with a score of 9 out of 10.
- The study and writing should be better focused; the authors still fall short in demonstrating the link between mood disorders and cognitive dysfunction in the present work.
Author Response
Thank you for the advice.
- It is checked and revised.
- With the cut-off point of 9/10, the sensitivity/specificity is low and not useful in defining the probable cognitive dysfunction. TCQ may still be able to help busy clinicians and MDD patients as well as their caretakers in identifying possible cognitive symptoms and facilitate further referral for detailed assessment of cognitive function as cognitive dysfunction is easily neglected. This idea is also added in Discussion.
- The link between mood disorders and cognitive dysfunction is elaborated more in Introduction.